# New Evidence on Regucalcin, Body Composition, and Walking Ability Adaptations to Multicomponent Exercise Training in Functionally Limited and Frail Older Adults

**DOI:** 10.3390/ijerph19010363

**Published:** 2021-12-30

**Authors:** Jorge Pérez-Gómez, Pedro C. Redondo, David Navarrete-Villanueva, Gabriel Lozano-Berges, Ignacio Ara, Marcela González-Gross, José A. Casajus, Germán Vicente-Rodríguez

**Affiliations:** 1HEME Research Group, Faculty of Sport Sciences, University of Extremadura, 10003 Cáceres, Spain; 2Department of Physiology, University of Extremadura, 10003 Cáceres, Spain; pcr@unex.es; 3Centro de Investigación Biomédica en Red de Fisiopatología de la Obesidad y Nutrición (CIBERObn), GENUD (Growth, Exercise, NUtrition and Development) Research Group, Faculty of Health and Sport Science (FCSD), Instituto Agroalimentario de Aragón-IA2- (CITA-Universidad de Zaragoza), 22001 Huesca, Spain; dnavarrete@unizar.es (D.N.-V.); glozano@unizar.es (G.L.-B.); joseant@unizar.es (J.A.C.); gervicen@unizar.es (G.V.-R.); 4Department of Physiatry and Nursing, University of Zaragoza, 50009 Zaragoza, Spain; 5GENUD Toledo Research Group, Department of Physical Activity and Sport Sciences, Universidad de Castilla-La Mancha, 45071 Toledo, Spain; Ignacio.Ara@uclm.es; 6CIBER of Frailty and Healthy Aging (CIBERFES), 28029 Madrid, Spain; 7ImFine Research Group, Department of Health and Human Performance, Universidad Politécnica de Madrid, 28040 Madrid, Spain; marcela.gonzalez.gross@upm.es; 8CIBER Physiopathology of Obesity and Nutrition (CIBEROBN), Instituto de Salud Carlos III, 28029 Madrid, Spain

**Keywords:** endurance training, human, physical exercise, physical fitness, strength training

## Abstract

Background: Regucalcin, or senescence marker protein-30 (SMP30), is a Ca^2+^-binding protein with multiple functions reported in the literature. Physical exercise has been shown to improve aging markers; nevertheless, SMP30 in humans has not been extensively researched. Older adults experience a decline in functional capacity and body composition. The purpose of this study was to examine the effects of a multicomponent training (MCT) program on SMP30 and its regulation of walking ability and body composition in functionally limited, frail, and pre-frail older adults. Methods: A total of 34 older adults (aged 80.3 ± 6.1 years) were divided into an intervention group (IG = 20) and control group (CG = 14). The IG performed a supervised MCT (strength, endurance, balance, coordination, and flexibility) program for 6 months, 3 days per week, whereas the CG continued their normal lives without any specific physical training. SMP30 was analyzed in plasma after 3 and 6 months of MCT, while some physical fitness variables (Timed Up and Go (TUG) and 6-min walk test (6MWT)) and body composition (fat mass and lean mass) were measured at baseline, as well as after 3 months and 6 months of MCT. Results: No significant changes were observed in SPM30 between the IG (877.5 a.u. to 940.5 a.u., respectively) and CG (790.4 a.u. to 763.8 a.u., respectively). Moreover, no SMP30 differences were found between groups after 3 and 6 months of MCT. The IG improved significantly in the 6MWT after 3 months (472.2 ± 84.2 m) compared to baseline (411.2 ± 75.2 m). The IG also significantly enhanced their TUG performance after 3 months (7.6 ± 1.6 s) and 6 months (7.3 ± 1.8 s) of training compared to baseline (9.3 ± 3.2 s) (all, *p* < 0.001). There were no significant differences in body composition between the IG and CG through the 6 months of MCT. Conclusions: The present study suggests that MCT did not change SMP30 levels from 3 to 6 months, where there were changes in neither walking ability nor body composition; however, MCT was effective in improving 6MWT and TUG performance from baseline to 3 months.

## 1. Introduction

Regucalcin, or senescence marker protein-30 (SMP30), is a calcium-binding protein with multiple physiological functions [1]. SMP30 participates as a suppressor protein of cell signaling in many cells, maintains calcium homeostasis [2], and is involved in brain calcium signaling, where the calcium accumulation can be related to brain toxicity [3]; SMP30 can also prevent apoptosis [2]. Low hepatic SMP30 can exacerbate nonalcoholic fatty liver disease and fibrosis in humans [4]. The overexpression of SMP30 contributes to glucose utilization and lipid production, while a deficit of SMP30 has been associated with deterioration of glucose tolerance and induction of hyperlipidemia by accumulation of triglycerides and total cholesterol in the liver of rats [4]. SMP30 has shown antitumor activity [5], and the development of SMP30 gene expression has been associated with preventive effects against the progression of cancer cells—SMP30 can reduce the progression of carcinogens and be useful in cancer treatment [6,7], and the expression of SMP30 in cancer cells is higher compared to normal tissue [8]. SMP30 also presents anti-inflammatory and antioxidant functions [9], and has been correlated with osteoporosis [8] and renal failure [10]. Thus, SMP30 can be an important molecule in metabolic disorders, maintaining calcium homeostasis and the functionality of other tissues against carcinogens and apoptosis. However, the aging process reduces the expression of the SMP30 [11,12]; as a consequence, its inhibitory effects have been observed to be debilitated with aging in rats [3]. However, despite the great potential of SMP30, to date only one study has been published about the effect of exercise on SMP30 in humans [13]. Pérez-Gómez et al. found that 12 weeks of whole-body vibration training was effective in improving SMP30 levels in postmenopausal women [13]. It is relevant to know whether any other types of exercise have positive effects on SMP30 levels.

Aging is associated with many physiological changes [14,15,16,17] that deteriorate physical fitness [18,19] and body composition, producing health issues [20,21]. Improving physical fitness is important for daily activities in older people [22]; in the same way, obesity—an excess of fat mass—constitutes a global pandemic, with negative consequences for the elderly [23]. Hence, regular physical exercise [24]—and specifically multicomponent training (MCT) programs [25]—constitutes an alternative to slow down the decline in physical fitness and body composition in the elderly. As physical exercise and SMP30 seem to have synergic effects on many health and aging markers, it is plausible to hypothesize that the effects of exercise may be mediated by the action of SMP30. Therefore, the aim of this study was to analyze whether regular MCT can improve SMP30 and its relationship with walking ability and body composition in the elderly.

## 2. Materials and Methods

### 2.1. Recruitment and Screening

The complete methodology of the MCT included within the framework of the EXERNET-Elder 3.0 project is described in detail elsewhere [26]. In brief, four primary care health centers and three nursing homes for non-dependent people in the city of Zaragoza collaborated in the recruitment. Medical doctors, nurses, and heads of residences encouraged potential candidates to participate, and performed a preliminary screening to volunteers for limited function and being at risk of frailty, based on the cutoffs of the Short Physical Performance Battery (SPPB) for frailty (4 ≥ SPPB points ≤ 6) and pre-frailty (7 ≥ SPPB score points ≤ 9) [27,28]. Frailty was evaluated based on the Clinical Frailty Scale [29] and the FRAIL Scale [30]. After that, researchers contacted eligible participants to undergo the initial interview to collect personal data. The exclusion criteria were being younger than 65 years, having been diagnosed with dementia and/or cancer, and/or being dependent or robust (SPPB score less than 6 or greater than 9). Body composition and walking ability were measured at baseline, 3 months and 6 months, while SMP30 levels were analyzed at 3 and 6 months.

### 2.2. Randomization and Allocation

The sample was divided by convenience to maximize training attendance. Those participants who were unable or unwilling to attend training sessions regularly for 6 months were directly included in the control group (CG). Intervention group (IG) allocation was assigned according to the participant’s place of residence, in order to facilitate their attendance at training sessions. This group (*n* = 20) performed supervised MCT for 6 months (24 weeks), while the CG (*n* = 14) followed their usual lifestyles and routine activities throughout the study.

Although it was not possible to blind exercise from non-exercise groups, no expectations of superiority of either exercise protocol were held by investigators, nor conveyed to participants; thus, the IG was in essence blind to a treatment hypothesis.

### 2.3. Multicomponent Training (MCT) Program

The characteristics of the MCT have been described previously [26]. Briefly, the IG performed three training sessions of 1 h duration, composed of two strength sessions (SSs) and one endurance session (ES). Sessions were separated by 48 h, and included 10 min of warm-up, 35–40 min of main exercise, and 10–15 min of cool-down. In the SSs, participants performed different exercises to improve the strength and power levels of the upper limbs, lower limbs, and trunk, in addition to static balance and functional performance of daily living activities (DLAs). The ESs required participants to perform exercises to increase cardiorespiratory fitness levels, dynamic balance, coordination, and motor skills.

### 2.4. Adherence and Motivation Strategy

Trainers registered the attendance of participants every session. Furthermore, a motivational strategy was used based on obtaining the maximum percentage of adherence to the MCT (number of sessions attended divided by the total number of programmed sessions). The three participants of each group with the highest levels of attendance received some sporting equipment as a reward. Moreover, doctors and nurses of health centers and heads of residences involved in the project encouraged participants to continue attending the training sessions, especially at the beginning.

### 2.5. Body Composition Measurements

Height was measured with a 2.10 m portable stadiometer with a 0.001 m error margin (SECA, Hamburg, Germany). A portable bioelectrical impedance analyzer with a 200 kg maximum capacity and a +/− 50 g error margin (TANITA BC 418-MA Tanita Corp., Tokyo, Japan) was used to assess the body weight (kilograms), and to estimate the percentage of body fat and fat-free mass. Before weighing, participants removed shoes and heavy clothes. Body mass index (BMI) was calculated using the accepted method (BMI = weight/height^2^; kg/m^2^).

### 2.6. Blood Collection

Venous blood sampling was performed according to the recommendations of the European Federation of Clinical Chemistry and Laboratory Medicine’s Working Group for Preanalytical Phase and the Latin American Working Group for Preanalytical Phase [31]. The blood collection was performed by direct venous puncture in the antecubital vein using a 21-gauge butterfly needle (BD Vacutainer Safety-Lok™, BD Biosciences, Australia). Blood samples for biochemical analysis were allowed to clot for at least 60 min, then centrifuged, and sera were collected.

### 2.7. Western Blotting

Samples of plasma were unfrozen, and proteins were immediately denatured under reducing conditions by mixing them with an equal volume of Laemmli buffer (2×, 10% DTT). Protein-containing samples were heated for 10 min at 70 °C, and subsequently frozen at 4 °C to complete protein denaturation. Protein samples were finally separated by 10% sodium dodecyl sulfate–polyacrylamide gel electrophoresis (SDS–PAGE). Western blotting was completed using nitrocellulose membranes and the specific anti-SMP30 antibody diluted 1:1000 in Tris-buffered saline with 0.1% Tween^®^ 20 detergent (TBST), and incubated for 1 h at room temperature. After using the appropriate HRP-conjugated secondary antibody (1:10,000 for 1 h), membranes were developed using SuperSignal solution and a C-DiGit device (LI-COR^®^, Lincoln, NE, USA). Membranes were re-probed with an anti-actin antibody (1:1000 for 1 h) in order to ensure that similar amounts of protein were loaded in each cell line. The samples were pooled together and loaded in the same gel by intercalating the pre-test and post-test samples.

### 2.8. Walking Ability

Gait performance was initially measured with the 6-min walk test (6MWT) and then the Timed Up and Go (TUG) test, with at least 5 min of recovery between the two tests. The 6MWT was performed once and determined the maximum distance, in meters, that the participants could walk as fast as possible within the 6-min time limit around a rectangular course of 46 m. At least two participants had to perform the test simultaneously. After the completion of the test, the distance was recorded [32]. The TUG test was performed twice, and the time was measured in seconds. A subject would rise from a chair, walk 3 m, turn around, walk back to the chair and sit down. The result of the best trial was noted in [33]. A chronometer was used to record the time taken to complete the test.

### 2.9. Ethical Committee

The protocol of this study was approved by the Ethics Committee of Clinical Research at the Alcorcón Foundation University Hospital (16/50), and followed the ethical guidelines of the Declaration of Helsinki, 1961, revised by Fortaleza (2013) [34], complying with the Spanish legislation and legal regulations on clinical research in humans (Law 14/2007 on biomedical research). The study was registered on Clinicaltrials.gov (NCT03831841). Participants received detailed information—both orally and in writing—about the purpose, procedures, benefits, risks and discomfort that might result from participation in the study. All subjects who voluntarily agreed to participate signed an informed consent form prior to their first evaluation.

### 2.10. Statistical Analysis

Statistical analyses were performed using the Statistical Package for the Social Sciences (SPSS) version 22.0 for Mac OS X (SPSS Inc., Chicago, IL, USA). Data were presented as the mean standard deviation. Most variables were not normally distributed using the Shapiro–Wilk test and, consequently, non-parametric statistics were applied.

At baseline, the Mann–Whitney test was used to check differences between groups for descriptive characteristics. Moreover, this test was performed to examine differences in SMP30 levels, walking ability, and body composition between the IG and CG. The Wilcoxon signed-rank test was performed to examine differences in SMP30 levels between the 3- and 6-month evaluations in both the IG and CG. The Friedman test was used to analyze walking ability and body composition differences between the baseline, 3-month, and 6-month evaluations in both groups. When the overall effect of the Friedman test was significant, Wilcoxon signed-rank tests corrected by the number of tests (Bonferroni correction for three measurements) were applied. Thus, the level of significance used in these Wilcoxon signed-rank tests was 0.0167. The statistical power of the Mann–Whitney tests was calculated for SMP30 after 6 months of MCT. In the present study, means and standard deviations in the IG (*n* = 20) and CG (*n* = 14) were 940.5 ± 543.3 a.u. and 763.8 ± 285.2 a.u., respectively. Thus, assuming 1-tailed testing and alpha = 0.05, the sample’s statistical power (1-beta) was 0.317. To conduct a more detailed analysis, effect size statistics using r were calculated by dividing the z-score by the square root of the total number of participants/observations [35]. The effect size for r can be small (0.1–0.3), medium (>0.3–0.5), or large (>0.5). The level of significance was set at *p* < 0.05.

## 3. Results

### 3.1. Descriptive Data

Descriptive characteristics of participants divided into the IG and CG are shown in Table 1. The IG was significantly taller than he CG (*p* = 0.009; r = −0.22). No significant age, weight, or BMI differences were found between groups (*p* > 0.05; r ranged from −0.04 to −0.47).

### 3.2. SMP30, Walking Ability, and Body Composition

Table 2 summarizes SMP30, walking ability, and body composition variables measured at baseline, and after 3 and 6 months of MCT in the IG and CG. No significant differences were found between groups in the SMP30 and walking ability variables at any evaluation (*p* > 0.05; r ranged from −0.04 to −0.33). Nevertheless, the IG had a lower body fat percentage and higher lean mass than the CG at baseline, and after 3 and 6 months of MCT (*p* < 0.05; r ranged from −0.45 to −0.59).

The IG showed a significant improvement in TUG performance from baseline up to 3 months and 6 months of MCT (*p* < 0.0167; r: −0.50 to −0.56, respectively; Table 2), and in 6MWT performance from baseline up to 3 months (*p* = 0.001; r: −0.58). The body fat percentage and lean mass of the IG did not significantly change during and after MCT (*p* > 0.05). There were no changes in the walking ability and body composition parameters of the CG (*p* > 0.05).

## 4. Discussion

The purpose of this study was to analyze whether MCT can improve SMP30, walking ability, and body composition in the elderly. It is known that the level of expression of SMP30 decreases with aging in animals [12] and humans [36], and this seemed to be the tendency in the CG (from 790.4 a.u. to 763.8 a.u.) of this study, while the IG showed a tendency towards increased SMP30 levels, from 877.5 a.u. to 940.5 a.u., although the increase was not significant. Recently, our research group found that the expression of SMP30 increased significantly after 12 weeks of whole-body vibration training [13]; two main reasons could explain the contradictory results between the two studies: Firstly, we analyzed the SMP30 levels when the participants had been training for 3 months, so the levels of SMP30 could be higher after the first 3 months of training, while the possibility to increase more—from 3 to 6 months—could be limited, as we observed in the previous study in postmenopausal women, where there was an increase (27.7%) in SMP30 measured in plasma from baseline up to 3 months of vibration training [13]. It would have been ideal to measure the SMP30 from baseline up to 3 months of MCT. Secondly, in this study it seems that the characteristics of the MCT (intensity, volume) were less effective than the whole-body vibration training used in the study cited above [13], because the MCT did not lead to any observed changes in body composition—specifically, fat mass—while in the previous study there was an increase in fat mass of 4.4% from baseline to 3 months [13].

With regard to the walking ability, a positive effect of MCT on functional and health outcomes for seniors has been shown [37], making this kind of training an alternative for elders to exercise and enhance their walking ability. The 6MWT and TUG are valid tests to measure functional capacity [38] and mobility [33] in older people, respectively. The IG improved significantly in the 6MWT (14%) from baseline up to 3 months of MCT, proving effective in enhancing the physical capacity of the participants; this improvement is consistent with previous studies [13]; for example, Pérez-Gómez et al. found an increase of 12.5% in 6MWT performance after 3 months of training in postmenopausal women. However, there were no improvements in 6MWT performance between 3 months and 6 months, so it could be the case that during this period the MCT was less effective, which could also justify the lack of significant improvement in other variables, such as SMP30, as explained above. In the same way, the improvement in TUG performance was observed from baseline up to 3 months, with no additional increase from 3 to 6 months. The improvement shown in our results is consistent with previous studies that also found improvement in TUG performance after 3 months of exercise [39]. This leads us to hypothesize that the dosage and training loads after 3 months in these programs should be reviewed and adjusted with greater precision in order to obtain further benefits after the adaptations observed in the first 3 months.

No improvements in body composition were observed with the MCT in the elderly; this is consistent with previous studies, where 3–8 months of MCT were not enough to induce changes in body composition, body fat mass, or lean body mass in the elderly [40,41]; thus, it is possible that this kind of training method is not as effective as others in changing body composition in older people, because other types of training—such as whole-body vibration—were more effective in decreasing fat mass (−2.2% and 4.4%) after 3 months of training in elderly participants [13,42], even after 6 months [43]. More research is needed in order to optimize the characteristics of the MCT in order to reduce body fat in the elderly.

There are some limitations of this study. Blood samples were not taken at baseline; unfortunately, there is not much research on this topic—this is only the second study to measure SMP30 in humans undergoing training, so more research is necessary in order to elucidate whether changes in SMP30 may also be related to changes in walking ability and/or body composition. Another limitation was that evaluations were not blinded, because the trainers of the intervention groups took part in them. The sample size could be increased in order to observe more changes. Another important limitation related to the MCT training is the fact that controlling some load variables—such as intensity—in all of the exercises involved in the MCT is difficult, so those variables were guided based on participants’ feelings and the guidance of the scientific literature. Due to participants’ availability, they were not randomized, which could also be considered a design limitation.

## 5. Conclusions

The present study showed that SMP30 levels did not change from 3 to 6 months; during this period there were changes in neither walking ability nor body composition. MCT was effective in improving 6MWT and TUG performance from baseline to 3 months, but seemed to be less effective from then on.

## Figures and Tables

**Table 1 ijerph-19-00363-t001:** Descriptive characteristics of participants assigned to control or intervention groups.

Variables	Control Group	Intervention Group
Age (year)	80.4 ± 5.9	81.0 ± 6.2
Weight (kg)	72.2 ± 15.5	77.6 ± 14.7
Height (cm)	153.3 ± 6.5 *	161.8 ± 8.3
BMI (kg/m^2^)	30.6 ± 4.9	29.7 ± 5.7

BMI: body mass index. * *p* < 0.05 between groups.

**Table 2 ijerph-19-00363-t002:** SMP30, walking ability, and body composition in elders assigned to control or multicomponent training program groups.

	**Control Group**				**Intervention Group**			
	**Baseline**	**3 Months**	**6 Months**	** *p* **	**Baseline**	**3 Months**	**6 Months**	** *p* **
SMP30 (a.u.)	-	790.4 ± 320.9	763.8 ± 285.2	0.730	-	877.5 ± 392.4	940.5 ± 543.3	0.526
TUG (s)	10.7 ± 4.9	9.3 ± 3.2	10.7 ± 7.0	0.846	9.3 ± 2.4 #$	7.6 ± 1.6	7.3 ± 1.8	0.005
6MWT (m)	368.0 ± 151.7	366.6 ± 146.9	361.1 ± 138.1	0.957	411.2 ± 75.2 #	472.2 ± 84.2	448.4 ± 108.5	0.004
Body fat (%)	42.5 ± 3.3 *	42.5 ± 3.3 *	42.1 ± 3.3 *	0.459	36.0 ± 5.0	36.1 ± 4.9	35.5 ± 5.1	0.336
Lean mass (kg)	40.4 ± 9.1 *	40.6 ± 10.0 *	40.6 ± 9.2 *	0.236	49.2 ± 8.9	49.5 ± 8.7	49.4 ± 8.9	0.689

SMP30: senescence marker protein-30; a.u.: arbitrary units; TUG: Timed Up and Go test; 6MWT: 6-min walk test. *p* < 0.05: * between groups at the same evaluation; # intragroup between baseline and 3 months; $ intragroup between baseline and 6 months.

## Data Availability

The average of the data from all subjects are included in the article.

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
