# Peer review of "New Evidence on Regucalcin, Body Composition, and Walking Ability Adaptations to Multicomponent Exercise Training in Functionally Limited and Frail Older Adults"

_ijerph, 2021, doi:10.3390/ijerph19010363_

Round 1

Reviewer 1 Report

I believe the authors tap into an important topic in sport science and health in older populations.

This randomized control trial aims to examine the effects of a multicomponent training program on SMP30 and its regulation 24 on walking ability and body composition in functional limited, frail and pre-frail older adults. This paper tries to explain in a positive way when looking at the physical exercise of older people and strategies to improve their health. I think some things need clarifying for the publication that will help in the overall interpretation and understanding of the results before being published within the scope of IJERPH.

Comment 1: The authors have presented a clear and well-written manuscript.

Material and Methods

Comment 2: The circadian rhythm influence physical assessments and blood collection. How did the authors control this variable?

Comment 3: From 112 to 120 lines, it is not a necessary paragraph. All procedures are specifically described in reference 26. Please add this reference and delete the lines.

Comment 4: The strength training protocol is not available in reference 31. The reference cited describes the protocol of venous blood sampling. Please add a specific reference. Can the authors describe the types of strength exercises (i.e. leg, extension, leg curl, curl biceps,…) used in the study?

Comment 5: The order of physical fitness assessments can influence the results. Can the authors describe the order of physical fitness assessments? The time recovery between physical fitness tests?

Comment 6: Table 1 – Please add descriptive characteristics of participants for body fat and lean.

Comment 7: Table 2 – The lean mass and body fat had significant changes across the study. The authors did not consider the body composition variables as adjustment variables when comparing the means of SMP30. Can the authors clarify this?

Comment 8: The intensity of training is an important variable that probably conditioned the acute and chronic response at the training in this study. Why the authors did not consider this variable in the discussion of their results?

Comment 9: Older people are a sensitive population to the acute response to physical exertion (i.e. heart rate, blood pressure,...). For participant health safety, how did the authors control for the acute response to exercise? By HRV? By RPE?

Comment 10: All participants were informed about the aim of the project? The written consent was obtained from their legal guardians before participation in the study? Please provide the reference.

Author Response

I believe the authors tap into an important topic in sport science and health in older populations.

This randomized control trial aims to examine the effects of a multicomponent training program on SMP30 and its regulation 24 on walking ability and body composition in functional limited, frail and pre-frail older adults. This paper tries to explain in a positive way when looking at the physical exercise of older people and strategies to improve their health. I think some things need clarifying for the publication that will help in the overall interpretation and understanding of the results before being published within the scope of IJERPH.

 Comment 1: The authors have presented a clear and well-written manuscript.

Thank you very much.

Material and Methods

Comment 2: The circadian rhythm influence physical assessments and blood collection. How did the authors control this variable?

Dear Reviewer, it is an interesting point, for that reason all physical tests and blood collection were measured at the same time on the different days, according to the standard procedures for normal blood sampling and analysis.

Comment 3: From 112 to 120 lines, it is not a necessary paragraph. All procedures are specifically described in reference 26. Please add this reference and delete the lines.

Done

Comment 4: The strength training protocol is not available in reference 31. The reference cited describes the protocol of venous blood sampling. Please add a specific reference. Can the authors describe the types of strength exercises (i.e. leg, extension, leg curl, curl biceps,…) used in the study?

Dear Reviewer, as you suggested we deleted paragraphs from 112 to 120. The specific types of strength exercises are described in reference 26.

Comment 5: The order of physical fitness assessments can influence the results. Can the authors describe the order of physical fitness assessments? The time recovery between physical fitness tests?

Thank you, it was not included, we added this information in the text: “Gait performances were measured in the first place with the 6-min walk test (6MWT) and then the time up and go (TUG), with at least 5 minutes of recovery between both tests”. Lines 162-164

Comment 6: Table 1 – Please add descriptive characteristics of participants for body fat and lean

It is included in table 2, to avoid the same data twice we believe it is better do not to duplicate it in table 1.

Comment 7: Table 2 – The lean mass and body fat had significant changes across the study. The authors did not consider the body composition variables as adjustment variables when comparing the means of SMP30. Can the authors clarify this?

This is a convenient observation, however since most of the variables were not normally distributed, the non-parametric test used did not allow for adjustment. Although, there was a baseline significant different that was maintained through the study, new studies with higher n could look into this issue in more detail.

Comment 8: The intensity of training is an important variable that probably conditioned the acute and chronic response at the training in this study. Why the authors did not consider this variable in the discussion of their results?

The intensity is one of the limitations included in the article, so we consider not to speculate about this variable in the discussion

Comment 9: Older people are a sensitive population to the acute response to physical exertion (i.e. heart rate, blood pressure,...). For participant health safety, how did the authors control for the acute response to exercise? By HRV? By RPE?

Unfortunately, we did not control it, but you are right, it is an aspect that we must take into account in future studies, thank you very much

Comment 10: All participants were informed about the aim of the project? The written consent was obtained from their legal guardians before participation in the study? Please provide the reference.

Of course, all this information is in the point 2.9. additionally, we sent to the Int. J. Environ. Res. Public Health a copy of the standard consent.

Dear reviewer, thank you very much for your comments. We think the quality of the article has increased with your suggestions.

Reviewer 2 Report

I read with great interest this work entitled New evidences on Regucalcin, body composition and walking ability adaptations to multicomponent exercise training in functional limited, and frailty older adults. If the form of the document is acceptable, it has a significant methodological bias. Indeed, the authors did not measure the baseline level of SMP30. However, this variable is their variable of interest.

If the form of the document is acceptable, it has a significant methodological bias. Indeed, the authors did not measure the baseline level of SMP30. Yet this variable is their variable of interest. Thus, the effects of exercise at 3 and 6 months compared to baseline cannot be investigated.

Thus, I suggest that authors present their work in the form of a brief report. Because the work lacks the methodological strength to be presented as an article.

Author Response

I read with great interest this work entitled New evidences on Regucalcin, body composition and walking ability adaptations to multicomponent exercise training in functional limited, and frailty older adults. If the form of the document is acceptable, it has a significant methodological bias. Indeed, the authors did not measure the baseline level of SMP30. However, this variable is their variable of interest.

If the form of the document is acceptable, it has a significant methodological bias. Indeed, the authors did not measure the baseline level of SMP30. Yet this variable is their variable of interest. Thus, the effects of exercise at 3 and 6 months compared to baseline cannot be investigated.

Thus, I suggest that authors present their work in the form of a brief report. Because the work lacks the methodological strength to be presented as an article.

Dear Reviewer, thank you for your comments. Yes, we do not have baseline data for SMP30 and it is a limitation included in the article, even with this limitation and due too there are no information about physical exercise and SMP30 in humans we consider it is really important to present this work as an article in order to include all study protocol that has been carried out, it will help researcher to improve future studies with all the information included in this article. On the other hand, to change from article to brief report unfortunately is not going to fix the limitation, beside the information, in this relevant topic, for readers or researchers will be limited.

Reviewer 3 Report

Thank you for the opportunity to review this paper. In general, the paper is well-written and presents novel evidence. Given that the data is from an existing overarching study/project, and given that the paper is relatively short, it could be considered to be reshaped as short communication paper.

The English is decent, but a proofread by a native speaker would be beneficial. Here my comments that will hopefully help the authors to improve the paper.

  • Title: I think it should be evidence, not evidences
  • Lines 33-38. Please report standard deviations in addition to mean values.
  • Line 62-63. Please spend 1-2 sentences to explain what this study (ref.13) found; this will help with establishing the rationale for the study
  • Was the sample size calculated a priori? If not, it is not a problem, in such case, can you provide a post-hoc calculation of statistical power?
  • The course for the 6-min walking test is unusual, can you provide a reference? The most usual is 5x10m rectangle.
  • Subheading 2.9 is somewhat repetitive of the information in lines 85-86; maybe include all ethics consideration in 2.1
  • Line 182. The symbol before SD is missing, at least in the PDF version of the paper?
  • Please include effect sizes for the analyses of differences.
  • Can you somehow reshape the table to be more readable? Now, some rows are larger than others.
  • You say in the methods section that you will assess correlations, but none are reported in the results.
  • On that note, I think you should assess the correlations between CHANGES in SMP30 and CHANGES in other variables.
  • You should stress possible allocation bias in the limitations.

Author Response

Thank you for the opportunity to review this paper. In general, the paper is well-written and presents novel evidence. Given that the data is from an existing overarching study/project, and given that the paper is relatively short, it could be considered to be reshaped as short communication paper.

The English is decent, but a proofread by a native speaker would be beneficial. Here my comments that will hopefully help the authors to improve the paper.

  • Title: I think it should be evidence, not evidences
  • Corrected
  • Lines 33-38. Please report standard deviations in addition to mean values.
  • Added
  • Line 62-63. Please spend 1-2 sentences to explain what this study (ref.13) found; this will help with establishing the rationale for the study
  • Good idea, thank you, we included: “Pérez-Gómez et al. found that 12 weeks of whole body vibration training was effective to improve SMP30 in postmenopausal women [13], it is relevant to know if any other type of exercise have positive effect on SMP30.”
  • Was the sample size calculated a priori? If not, it is not a problem, in such case, can you provide a post-hoc calculation of statistical power?

Thank you for this observation. The information has been added as follows:

The statistical power for Mann-Whitney test was calculated for SMP30 at 6-month of MCT. In the present study, means and standard deviation in IG (n=20) and CG (n=14) were 940.5±543.3 a.u. and 763.8±285.2 a.u., respectively. Thus, assuming 1-tail testing and alpha=0.05, the sample the statistical power (1-beta) was 0.317. To conduct a more detailed analysis, effect size statistics using r were calculated by dividing z-score by the square root of the total number of participants/observations [35]. The effect size for r can be small (0.1–0.3), medium (>0.3–0.5) or large (>0.5).

  • The course for the 6-min walking test is unusual, can you provide a reference? The most usual is 5x10m rectangle.

Thank you for the comment, there are different tests, however, in older persons it is quite common also this 6MWT which is included in the senior battery test (1), that is the battery used for the EXERNET-Elderly cohort from 2008 for which we have also proposed reference values in spain (2). So in order to continue with the same measurements in the studies related to this cohort we keep this test.

  1. Rikli, R.; Jones, C. Senior Fitness Test Manual, 2nd ed.; Tocco, A., Maurer, K., Cox, K., Shea, B., Feeney, J., Huls, S., Eds.; Human Kinetics: Champaign, IL, USA, 2002; ISBN 978-1-4504-1118-9.
  2. Physical fitness levels among independent non-institutionalized Spanish elderly: the elderly EXERNET multi-center study. Pedrero-Chamizo R, Gómez-Cabello A, Delgado S, Rodríguez-Llarena S, Rodríguez-Marroyo JA, Cabanillas E, Meléndez A, Vicente-Rodríguez G, Aznar S, Villa G, Espino L, Gusi N, Casajus JA, Ara I, González-Gross M; EXERNET Study Group. Arch Gerontol Geriatr (IF: 3.25; Q3). 2012 Sep-Oct;55(2):406-16. doi: 10.1016/j.archger.2012.02.004. Epub 2012 Mar 16.
  • Subheading 2.9 is somewhat repetitive of the information in lines 85-86; maybe include all ethics consideration in 2.1

Yes, it is repetitive, we deleted “, in which volunteers received detailed information of study and signed the informed consent” in lines 88-89

  • Line 182. The symbol before SD is missing, at least in the PDF version of the paper?
  • Sorry, we do not see any missed symbol in line 182
  • Please include effect sizes for the analyses of differences.

Effect size using r had been already calculated. This value was reported along Results section. Furthermore, the following information was detailed in Statistical Analyses section:

Effect size statistics using r were calculated by dividing z-score by the square root of the total number of participants/observations [35]. The effect size for r can be small (0.1–0.3), medium (>0.3–0.5) or large (>0.5).

  • Can you somehow reshape the table to be more readable? Now, some rows are larger than others.
  • Corrected
  • You say in the methods section that you will assess correlations, but none are reported in the results.
  • See next comment
  • On that note, I think you should assess the correlations between CHANGES in SMP30 and CHANGES in other variables.

It was a mistake, we deleted assess correlations in the results section.

You should stress possible allocation bias in the limitations.

We added: “Due to participants’ availability they were not randomized and it can be considering a design limitation”.

Dear reviewer, thank you very much for your comments. We think the quality of the article has increased with your suggestions.

Round 2

Reviewer 1 Report

The authors responded to all my comments.

I do not have further questions.

Thank you.

Reviewer 2 Report

No comment

Reviewer 3 Report

Thank you for addressing my concerns thoroughly. I have no further objections.